# Patent Protection Policy and Firms' Green Technology Innovation: Mediating Roles of Open Innovation and Human Capital

**Dong Chen**  **and Shi Chen** * 

School of Economics, Southwestern University of Finance and Economics, Chengdu 611130, China; 120020104012@smail.swufe.edu.cn
* Correspondence: chenshi@swufe.edu.cn

**Abstract:** Green innovations such as renewable energy technologies and cleaner process modifications are important technical routes and critical directions for reducing carbon emissions from industrial production processes. This study examines the impact of intellectual property protection on green technology innovation, constructing a progressive difference-in-differences model using 849 listed manufacturing firms panel data from 2007 to 2019 and taking the Chinese Intellectual Property Rights model cities as a quasi-natural experiment. Our study finds that the pilot policy significantly enhances corporate green innovation. When considering heterogeneity, the policy treatment effect is more remarkable for large firms, state-owned enterprises, and industries where technology can be easily imitated. Moreover, the mediating effect shows that the policy promotes green innovation by encouraging firms' research and development cooperation and increasing human capital levels. This study proposes that policymakers should reinforce intellectual property protection, encourage companies to be better and bigger, and emphasize the intermediary function of open innovation and human capital in green technology innovation.

**Keywords:** intellectual property rights; green technology innovation; open innovation; human capital; sustainable development

## 1. Introduction

Green technology innovation refers to the improvement of products or processes to reduce environmental burdens or achieve sustainable development goals [1], such as renewable energy technologies and cleaner process modifications. Green technology innovation has a significant double externality problem [1], and it is critical in reconciling environmental protection and economic growth. Green technology innovation can alleviate enterprises' ecological pollution issues, prompt enterprises to produce green products, and enhance market competitiveness. This win-win potential has caught the attention of policymakers, who are allocating increasing amounts of public funds to stimulate the creation and diffusion of clean and sustainable technologies. For example, the European Commission has developed a low-carbon economy roadmap to achieve emission reduction targets by 2050 [2]. Similarly, China has committed to reaching peak carbon by 2030 and carbon neutrality by 2060. Against the backdrop of increasing international emphasis on green development models, green technology innovation, which is a critical driver for accelerating green energy development, has made significant progress in China. According to the Global Green and Low Carbon Technology Patent Statistical Analysis Report (2023) released by the China National Intellectual Property Administration, China has authorized a total of 206,000 green and low-carbon patents from 2016 to 2022, accounting for 36.8% of the global green and low-carbon patent authorization.

What are the drivers of the dramatic increase in green patents in the manufacturing sector of China? A common factor, "the Porter hypothesis", states that environmental regu-

lation generates compliance costs and innovation offsets [3]. While negatively impacting green product innovation in the short run, stringent environmental regulatory policies drive China's green technology innovation and economic growth in the long run [4]. Furthermore, external factors, like China's aggressive green finance policy and the rapid expansion of the digital economy, and internal factors, such as firms' organizational capabilities, can affect corporate green technology innovation. However, an essential crucial element that has attracted limited focus in the literature is China's improving Intellectual Property Rights (IPR) regime, where more robust IPR protection, in general, can significantly and positively affect technological innovation [5–7]. Hu and Jefferson [8] reveal that the changes in China's patent law are one of the primary reasons for the geometric growth of patent applications since the mid-1980s. To further strengthen the protection of IPR, the Chinese government has promulgated several policies, such as the "Outline for the Construction of a Strong Intellectual Property State (2021–2035)".

Has China's improving IPR protection regime promoted green innovation? In this paper, our research objective is to provide empirical evidence and references for relevant policymakers by examining the impact and mechanism of China's IPR protection policies on firms' green technological innovation process. We create a dataset including a sample of 849 listed manufacturing firms in 230 cities in China from 2007 to 2019 for analysis. Regarding the empirical strategy, this study adopts a progressive difference-in-differences (DID) method for benchmark regression estimation. For consistency, we deal with possible endogeneity by performing a series of regressions using new estimation methods, alternative indicators, and sample data. To account for heterogeneity, we explore the changes in the Chinese IPR model city pilot policy on green technology innovation under different firm- and industry-level characteristics. Finally, using a mediating effects model, we analyze two primary mechanisms to examine how the pilot policy affects firms' green innovation.

Our research findings are as follows. First, the Chinese IPR model city pilot policy encourages enterprises to apply for more green patents. IPR protection policies play a continuous and long-term role in promoting green technology innovation. Second, there is heterogeneity in the impact of IPR protection on green technology innovation. The policy treatment effect tends to be more remarkable in enterprises with a larger size, state-owned character, and technology-reversing industries. Third, the pilot policy's promotion of green innovation in firms can be obtained through open innovation and human capital. Enhanced IP protection will, on the one hand, encourage firms to collaborate with external organizations in research and development (R&D); on the other hand, it will also improve the skill composition of firms' human capital and increase the share of scientific and technological personnel.

This study makes several contributions to the literature. First, to our knowledge, this is one of the first studies to examine the relationship between firm-level IP protection and green technology innovation by using the Chinese IPR model city pilot policy as a quasi-natural experiment. We confirm a new driving force from the perspective of IP protection for green technology innovation research. Second, we demonstrate that the IPR policies' promotion affects enterprises' green technology innovation in the long term and continuously. We use a mediating effects model to investigate the intermediary function of external R&D cooperation and internal human capital optimization. Our work enriches IP protection literature and expands intellectual property's role in corporate green technology innovation.

The remainder of this article is as follows. Section 2 reviews the relevant literature, describes the policy context, and analyses the mechanisms by which IP policies affect firms' green innovation. Section 3 introduces the estimation strategy, variables, and data. Section 4 analyses the results of the baseline regression and the robustness tests. Section 5 implements heterogeneity analyses and verifies the mechanisms. Finally, the last section concludes the work.

## 2. Institutional Background and Literature Review

### 2.1. The Chinese IPR Model City Pilot Policy

Since its introduction in 1984, the Chinese Patent Law has undergone four amendments, such as extending the patent protection period for inventions to 20 years. With the subsequent enactment of more IPR protection policies, China has made major progress in IPR protection, and the number of patents has rapidly risen [8]. However, the misuse of technological inventions, product piracy, and other IPR infringements often occur. These behaviors discourage companies from R&D and affect foreign direct investment [9].

To further improve the IP protection system, China issued the "Outline of National Intellectual Property Strategy" in 2008, requiring "in-depth pilot work on various types of intellectual property". The Chinese government conducted IPR pilot work at the city level in 2012, 2013, 2015, 2016, 2018, and 2019, and selected 77 national IPR model cities/districts in batches. The construction tasks of model cities mainly include four aspects:

(a) Increasing government investment. The pilot cities should incorporate IPR work into the local government's annual assessment indicators. Furthermore, the government should increase funding, such as increasing the financial investment in IPR.
(b) Strengthen the construction of IPR laws and regulations. The pilot cities should formulate or revise local regulations on IPR.
(c) Promote the administrative enforcement of IPR. Pilot cities should actively perform administrative enforcement of patents, rights protection assistance, and infringement reporting complaints, among others.
(d) Optimize the innovation environment. The pilot cities shall publicize and report on the protection of IPR through mainstream media, portals, information platforms, and other channels to create an atmosphere for IP protection.

### 2.2. Literature Review and Hypothesis

In the literature, there is extensive research regarding the impact of IP protection on technological innovation. Most scholars believe that stronger IPR can promote technological innovation [10]. The reason for this belief is that patent law reduces the firms' spillovers of R&D by guaranteeing monopoly gains to inventors [11], which further incentivizes firms to invest in R&D [12]. Specifically, IPR protection promotes technological innovation in three ways. First, IPR protection can reduce R&D spillovers and ensure the benefits of innovation to companies. There is an externality problem for corporate R&D activities, where competitors can reap the economic benefits of innovation for free by imitating or illegally stealing patented technologies [13]. Patent laws can confer exclusivity on innovations, and companies can continuously benefit through patent licensing or technology monopolies. IP protection will significantly reduce free-riding and, thus, stimulate corporate technology innovation. Second, IPR protection can ease information asymmetry and attract external investment. A company with stronger IP protection will be more willing to disclose innovation information to external shareholders and creditors, and it will be easier for the company to obtain external debt [7]. Third, from the perspective of international technology transfer, developing regions with higher IP protection can attract foreign direct investment. When local IPR protection is weak, multinational enterprises' R&D activities and technology applications are mainly conducted within the enterprise [14]. With more vital local IPR protection, multinational enterprises can earn higher returns through patent technology monopoly, enhancing developing countries' foreign direct investment and technology level [9].

As part of the innovation output, green technologies are likely to be affected by IPR protection. Based on the above analyses, recent literature has similarly investigated the relationship between IP protection and green innovation from the perspectives of R&D activities, external investment, and technology transfer. For example, Schaefer [15] argues that stronger IP protection will promote firms to actively develop green technologies if green technologies are more productive than other technologies. Cao et al. [16] find that IP protection can promote green innovation by boosting firms' investment in R&D

and attracting foreign investment entry. Dussaux et al. [17] find that strengthening IP protection can accelerate the international transfer of low-carbon technologies, such as solar photovoltaic and wind power. These results suggest that IP strength positively affects firms' green patent production [18]. In addition, a considerable body of literature on sustainable development emphasizes the critical role of IP protection. For instance, a study by Jiang et al. [19] found that the Chinese IPR model city pilot policy reduced carbon emissions in cities by promoting green technological innovations. Liu and Zhong [20] highlight the importance of IP protection in spurring technological innovations for environmentally friendly resource extraction. The Chinese IPR model cities pilot, as a representative policy initiative to enhance patent protection, will likely impact firms' green innovation. From the literature, we propose a hypothesis as follows:

**H1.** *The Chinese IPR model city pilot policy can accelerate enterprise green technology innovation.*

How does the Chinese IPR model city pilot policy affect firms' green technology innovation processes? The mechanism by which intellectual property protection affects firms' green innovation may differ from non-green innovation. According to Barbieri et al. [21], compared to non-green technologies, green technologies are more innovative, complex, and sustainable, requiring a higher level of skill, more diverse knowledge, and a unique combination of expertise in the technology development process. Hence, practitioners and scholarly literature advocate cross-industry collaborations to advance green technologies [22,23]. In this paper, we argue that the policy can promote firms' green innovation by encouraging R&D cooperation and increasing human capital levels. When internal knowledge is insufficient to support a green technology innovation, companies will acquire critical knowledge, skills, and resources from other companies, external suppliers, and universities through collaborative R&D [24]. Cooperative R&D is one of the determinants driving green technology innovation [25].

However, there is a paradox in the relationship between patent protection and open innovation [26]. Studies suggesting a positive correlation argue that firms tend to engage external partners when they can hinder technology spillovers through the patent system. For example, Cassiman and Veugelers [27] argue that firms in external collaborations often use patents to limit the dispersal of crucial knowledge to external collaborators. Conversely, studies proposing a negative correlation underscore that the exclusivity of patent protection reduces the effectiveness of cooperative R&D and diminishes the attractiveness of external partnerships. Emphasizing the protection of a company's proprietary information may complicate collaboration with external parties [28]. While still a topic of debate within the literature, this paper posits that stronger protection of IPR has significantly increased companies' willingness to engage in collaborative R&D with external industrial organizations. One reason for this phenomenon is that a robust IP legal system can reduce the R&D spillovers of a firm [11]. Companies gain access to critical knowledge and technology when working with external organizations and avoid imitation or infringement of internal technology [29]. According to Roh et al. [30], with open innovation as the mediator, IPR can significantly affect a firm's green process and product innovation. According to the above analyses, we propose the following hypothesis:

**H2.** *The Chinese IPR model city pilot policy can promote green innovation by encouraging firms' R&D cooperation.*

With the emergence of many new technologies and product updates, companies must invest more in innovation, especially in future-oriented green technologies, to maintain the industry's competitiveness and shorten product life cycles [31]. Thus, companies will tend to rely on a highly skilled workforce in upgrading their production technology. The Chinese IPR model city pilot policy is critical in bolstering the market supply of human resources and stimulating the demand for high-level talent within firms. On the supply side, local governments have increased financial support for the training and recruiting of

domestic and foreign technicians to promote the pilot policy. Consequently, the associated search and matching costs for firms to acquire human capital within the IPR pilot cities have been notably diminished. Additionally, established studies, such as Naghavi and Strozzi [32], underscore the role of IPR protection in attracting the return of international migrants by fostering a supportive, protective environment and innovative climate, thus driving domestic technological advancement.

From the demand perspective, the IPR regime empowers firms to derive monopoly rents or excess profits when innovating, by granting temporary exclusionary rights to patent owners [33]. Enhanced patent protection motivates firms to increase their investment in R&D, aiming to secure a larger market share and achieve competitive advantages [34], thereby highlighting the critical importance of R&D personnel engaged in technology development. Consequently, firms take a more proactive approach in cultivating and attracting technological talent, as seen in examples such as Huawei, which employs scientists at competitive salaries globally. Research by Melero et al. also attests that patent protection effectively reduces the mobility of inventors within firms, with an additional patent granted decreasing the likelihood of changing employers by 23% on average [35]. Therefore, we propose the following hypothesis.

**H3.** *The Chinese IPR model city pilot policy can promote green innovation by increasing firms' human capital levels.*

## 3. Empirical Strategy and Data

### 3.1. Methodology

This article assesses the effect of IPR policies on green technology innovation by conducting China's IPR model city pilot policy as a quasi-natural experiment. We use a progressive DID specification to assess the relationship between the Chinese IPR model city pilot policy and green technology innovation, with a specific regression setup referring to Beck et al. [36] and Tang et al. [37]:

$$Ginvent_{it} = \alpha + \beta Treat_i \times Post_t + \sum \gamma_j X_{it} + \delta_i + \theta_t + \varepsilon_{it} \tag{1}$$

where the dependent variable $Ginvent_{it}$ represents the level of green technology innovation of firm $i$ in year $t$. $Treat_i$ is a dummy variable for the Chinese IPR model city pilot policy, with a value of 1 for the city where the company locates in the pilot region and 0 for the opposite. $Post_t$ is a dummy variable before and after the policy, taking a value of 1 after the start of the pilot and 0 in other years. $X_{it}$ is a set of control variables that affect a company's green technology innovation, $\delta_i$ and $\theta_t$ represent firm- and year-fixed effects, respectively, and $\varepsilon_{it}$ is the error term. In model (1), we use the number of green invention applications by firm $i$ in year $t$ to measure the company's green innovation. The estimated coefficient $\beta$ of $Treat_i \times Post_t$ is our focus. A positive and significant $\beta$ indicates that the Chinese IPR model city pilot policy positively impacts companies' green technology innovation. In addition, we estimate Equation (1) using the standard errors of clustering at the city level.

### 3.2. Variable Measurement

#### 3.2.1. Dependent Variable

The dependent variable in model (1) is firms' green innovation capabilities, which is measured referencing general practice in the literature using the natural logarithm of the number of firms' green invention patent applications plus 1 (*Ginvent*) [38]. According to Du et al. [39], green patents most directly reflect the output of firms' green technology innovation activities, which can be quantified and categorized into technologies. Furthermore, various uncertainties are associated with patents from application to acceptance (e.g., long examination cycles, bureaucracy). Patent applications are a more accurate measure of current technological innovation activities compared with patents granted [40]. In addition, green inventions with a higher quality of innovation are more representative compared to

green utility models. For robustness testing, we also employ the number of green invention patents granted.

From the descriptive statistics in Table 1, we see the mean value of the number of green invention patent applications for the sampled firms, which is 0.6022. This result indicates a relatively low number of green patent applications among listed companies in China's manufacturing industry.

**Table 1.** Summary statistics (N = 9642).

| Variables | Mean | Std. Dev | Min | Max | Quantile | | |
|---|---|---|---|---|---|---|---|
| | | | | | 50% | 75% | 95% |
| *Ginvent* | 0.6022 | 0.9498 | 0.0000 | 4.2485 | 0.0000 | 1.0986 | 2.7081 |
| *Treat* | 0.5707 | 0.4950 | 0.0000 | 1.0000 | 1.0000 | 1.0000 | 1.0000 |
| *Post* | 0.3613 | 0.4804 | 0.0000 | 1.0000 | 0.0000 | 1.0000 | 1.0000 |
| *Size* | 3.5889 | 1.1474 | 1.4110 | 7.0901 | 3.4657 | 4.2485 | 5.7991 |
| *Mature* | 2.7217 | 0.3914 | 1.3863 | 3.4340 | 2.7726 | 2.9957 | 3.2581 |
| *Lev* | 0.3992 | 0.1898 | 0.0482 | 0.8240 | 0.3972 | 0.5458 | 0.7090 |
| *Share* | 0.3368 | 0.1390 | 0.0877 | 0.7215 | 0.3192 | 0.4258 | 0.5951 |
| *lnManager* | 1.9704 | 0.2915 | 1.3863 | 2.7726 | 1.9459 | 2.1972 | 2.4849 |
| *lnGov* | 0.2414 | 0.3498 | 0.0000 | 2.1342 | 0.1115 | 0.2738 | 0.9821 |

### 3.2.2. Core Independent Variables

We treat the pilot policy for the Chinese IPR model city pilot policy as a quasi-natural experiment. The interaction term between the dummy variables of city type and policy implementation time, *Treat* × *Post*, indicates the treatment effect of the IPR policy. We set the model cities *Treat* to 1 as the treatment group and non-model cities to 0 as the control group. The time dummy variable, *Post*, before and after the pilot policy, is set to 0 and 1, respectively. The Chinese IPR model cities list is from the China National Intellectual Property Administration. In 2012, 23 cities became the first IPR model cities, such as Chengdu and Shenzhen. Subsequently, 18, 12, 11, 6, and 7 pilot cities were added in 2013, 2015, 2016, 2018, and 2019, respectively. Considering the lag in policy implementation, only four batches of pilot cities (i.e., 2012, 2013, 2015, and 2016) are set up as treatment groups in this study. Table 1 shows the mean value of the variable for *Treat* is 0.5707, indicating that 57.07% of the sample belongs to the treatment group and 42.93% belongs to the control group.

### 3.2.3. Control Variables

Based on the drivers that influence companies' adoption of green innovation [31], we combined the studies by Tang et al. [37] and Zhou et al. [41] and selected the following control variables that potentially influence firms' green innovation. (a) Firm size (*Size*). Firm size affects the success of innovation, and larger firms tend to adopt more green technologies [42]. We use the natural logarithm of a firm's total assets to measure firm size. (b) Maturity (*Mature*). Companies that have been established for an extended time are likely to have accumulated relevant R&D experience and human capital and have a high innovation level. In the analysis, we use firm age to control the effects of a firm's maturity. (c) Corporate leverage (*Lev*). A reasonable level of debt facilitates companies to raise funds to engage in R&D for new products and green technologies. (d) Equity structure and management level (*lnManager*). Different shareholding structures and levels of management may influence whether a firm adopts an environmental innovation strategy [43]. We measure the corporate governance level by the percentage shares held by the largest shareholder (*Share*) and the number of senior executives. (e) Government subsidies (*lnGov*). Tax incentives and financial subsidies from the government sector in energy and the environment can vastly reduce companies' R&D costs and, thus, induce green innovation [44]. We collected data on government subsidies to control the impacts of the public sector. Table 1 shows that most of the standard deviations for the control variables are below 1, suggesting minimal variability in their changes. Additionally, examining

the correlation matrix depicted in Table 2 reveals small correlation coefficients among the variables, indicating a lack of significant multicollinearity issues.

**Table 2.** Correlation matrix.

| Variables | Ginvent | Treat×Post | Size | Mature | Lev | Share | lnManager |
|---|---|---|---|---|---|---|---|
| Treat × Post | 0.2633 *** | | | | | | |
| Size | 0.4439 *** | 0.1427 *** | | | | | |
| Mature | 0.1704 *** | 0.2181 *** | 0.3332 *** | | | | |
| Lev | 0.2274 *** | −0.0124 | 0.5026 *** | 0.1916 *** | | | |
| Share | −0.0664 *** | −0.0813 *** | 0.0770 *** | −0.1579 *** | −0.0106 | | |
| lnManager | 0.1113 *** | −0.0465 *** | 0.2471 *** | 0.0225 ** | 0.1210 *** | −0.0351 *** | |
| lnGov | 0.4602 *** | 0.1461 *** | 0.6315 *** | 0.2211 *** | 0.3040 *** | 0.0175 * | 0.1917 *** |

Note: *, ** and *** indicate significance at the 10%, 5%, and 1% levels, respectively.

*3.3. Data*

This study analyzes green patents and firm-level data of Chinese A-share listed companies from 2007 to 2019. The main reasons for choosing the sample period 2007–2019 in this study are as follows: first, to avoid the effect of the change in government grant guidelines in 2006 and the large number of missing values in the pre-2007 sample, and second, to mitigate the economic disruption caused by COVID-19. The green patent data utilized in this research is sourced from the Chinese Research Data Service (CNRDS) platform, encompassing a fusion of Chinese patent data and the green patent classification number standard, the International Patent Classification Green Inventory, issued by the World Intellectual Property Office. This dataset is widely employed across environmental studies to gauge the extent of green innovation within Chinese enterprises [45,46]. Listed companies' financial data are collected from the China Stock Market & Accounting Research (CSMAR) database. Corporate human capital data, such as the number and proportion of technology employees, are from the RESSET database.

As green technology innovation is mainly concentrated in the manufacturing sector, we have excluded service sectors such as finance and real estate and retained only the listed companies in the industrial sector. We winsorized the continuous variables at the 1% level to avoid the impacts of extreme values. Finally, we obtained unbalanced panel data for 849 listed manufacturing companies in 230 cities after excluding cities with incomplete data.

## 4. Empirical Results

*4.1. Time Trend Tests and Dynamic Effects*

The DID model's premise is that the treatment and control groups have the same change trend before the treatment. We adopt the event analysis approach to investigate the dynamic effects of the Chinese IPR model city pilot policy with the following model setup [47]:

$$Ginvent_{it} = \alpha + \beta_1 D_{it}^{-5} + \beta_2 D_{it}^{-4} + \cdots + \beta_6 D_{it}^0 + \cdots + \beta_{13} D_{it}^{+7} + \sum \gamma_j X_{it} + \delta_i + \theta_t + \varepsilon_{it} \quad (2)$$

The study period for this paper spans from 2007 to 2019, while the policy introduction year of the first IPR model cities is 2012. As a result, this paper assesses the influence of patent protection on firms' green innovation during the first five years before the policy's introduction (2007–2011) and the subsequent seven years after that (2013–2019). Based on model (1), we introduce 14 dummy variables. $D^{-j} = Treat \times Post^{-j}$ equals one for firms in the $j$th year before the IPR pilot, while $D^{+j}$ equals one for firms in the $j$th year after the IPR pilot.

Figure 1 shows that the coefficients are insignificant before the policy is introduced, indicating no significant difference between the treatment and control groups. In addition, the estimated coefficients after the policy pilot are significantly positive from the second year onwards and peak in the sixth year. These findings suggest that, although with a lag, the Chinese IPR model city pilot policy can positively impact green innovation both in the short and long term.

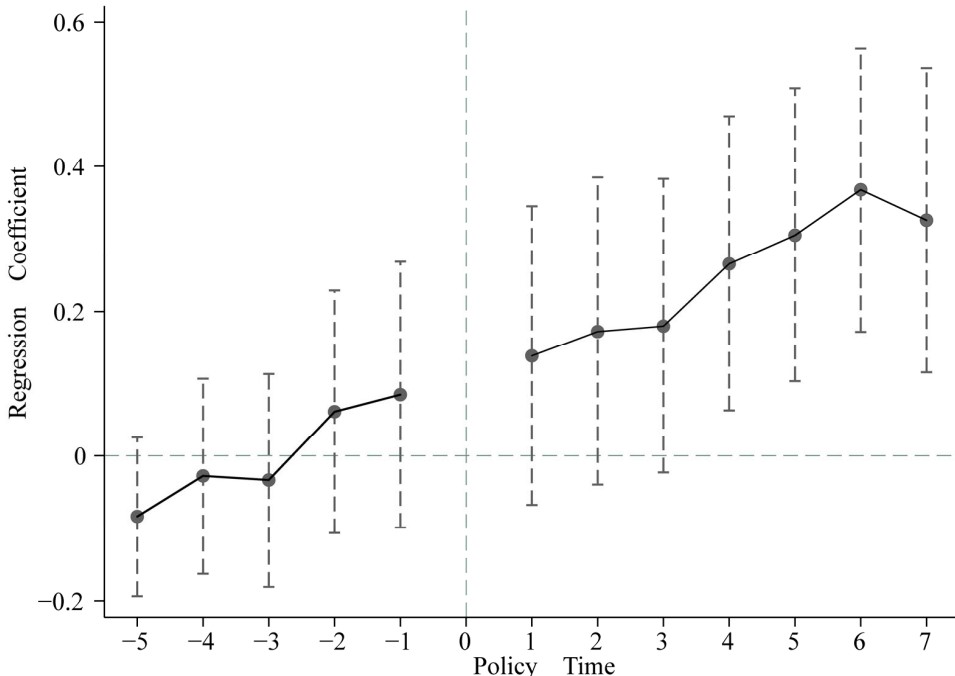

**Figure 1.** Time trend test.

*4.2. Baseline Regression Results*

Table 3 reports the regression results of the Chinese IPR model city pilot policy on green technology innovation by China's listed manufacturing companies, controlling for firm- and year-fixed effects in each column. Column (1) considers only the effect of the IPR pilot policy on firms' green innovation. Columns (2)–(5) are the regression results obtained by adding firm characteristics, business performance, management level, and government subsidies.

**Table 3.** Baseline regression results.

| Variables | (1) | (2) | (3) | (4) | (5) |
|---|---|---|---|---|---|
| | *Ginvent* | *Ginvent* | *Ginvent* | *Ginvent* | *Ginvent* |
| $Treat \times Post$ | 0.1600 *** | 0.1562 *** | 0.1593 *** | 0.1561 *** | 0.1485 *** |
| | (0.0441) | (0.0414) | (0.0414) | (0.0418) | (0.0389) |
| *Size* | | 0.3862 *** | 0.3996 *** | 0.4026 *** | 0.3259 *** |
| | | (0.0433) | (0.0449) | (0.0438) | (0.0412) |
| *Mature* | | −0.1233 | −0.0966 | −0.1285 | −0.0686 |
| | | (0.1114) | (0.1124) | (0.1113) | (0.1058) |
| *Lev* | | | −0.1675 * | −0.1730 * | −0.1811 ** |
| | | | (0.0879) | (0.0883) | (0.0837) |
| *Share* | | | | −0.2773 | −0.3245 * |
| | | | | (0.1956) | (0.1846) |
| *lnManager* | | | | −0.0302 | −0.0166 |
| | | | | (0.0498) | (0.0486) |
| *lnGov* | | | | | 0.4246 *** |
| | | | | | (0.0582) |
| Firm FE | Yes | Yes | Yes | Yes | Yes |
| Year FE | Yes | Yes | Yes | Yes | Yes |
| Constant | 0.5444 *** | −0.5047 * | −0.5597 * | −0.3273 | −0.3223 |
| | (0.0159) | (0.3015) | (0.3041) | (0.3261) | (0.3296) |
| $N$ | 0.6555 | 0.6751 | 0.6753 | 0.6756 | 0.6824 |
| $R^2$ | 9642 | 9642 | 9642 | 9642 | 9642 |

Notes: Robust standard errors for clustering to the city level are in parentheses. The *, **, and *** indicate significance at the 10%, 5%, and 1% levels, respectively.

The results show that the regression coefficient of *Treat* $\times$ *Post* is significantly positive at the 1% significance level in all cases. It suggests the Chinese IPR model city pilot policy has promoted enterprises' green technology innovation behaviors. In economic terms, the policy has resulted in a 16.01% ($= e^{0.1485} - 1$) increase in green invention patents relative to the level of non-IP pilot cities. In Column (5), the coefficients on both firm size (*Size*) and government subsidies (*lnGov*) are significantly positive at the 1% level. These results suggest that firm size and government subsidies positively influence the level of firms' green technology innovation.

*4.3. Robustness Checks*

4.3.1. PSM-DID Estimation

Because regions with better IPR development may be more likely to be selected as pilot cities by the China National Intellectual Property Administration, the nonrandom nature of the sample selection will lead to biased regression results. We use the propensity score matching difference-in-differences (PSM-DID) model for robustness testing to address the endogeneity problem caused by sample selection bias. First, we choose corporate characteristics, including firm size (*Size*), age (*Mature*), debts (*Lev*), equity structure (*Share*), number of executives (*lnManager*), and government subsidies (*lnGov*) as matching variables. Next, we refer to the method of Böckerman and Ilmakunnas [48] for period-by-period matching. Subsequently, the PSM samples are used for a progressive DID model for estimation. Column (1) of Table 4 reports the calculated results of the PSM-DID model. The coefficient of *Treat* $\times$ *Post* remains significantly positive and consistent with the baseline regression results. The finding suggests that after accounting for sample selection bias, the Chinese IPR model city pilot policy still significantly affects green innovation.

**Table 4.** Robustness tests.

| Variables | PSM-DID | Alternative Variables | | Change Sample | |
|---|---|---|---|---|---|
| | (1) | (2) | (3) | (4) | (5) |
| | *Ginvent* | *Ginvent_g* | *Gutility* | *Ginvent* | *Ginvent* |
| *Treat* $\times$ *Post* | 0.1451 *** | 0.0996 *** | 0.0843 ** | 0.1538 *** | 0.0984 *** |
| | (0.0388) | (0.0251) | (0.0331) | (0.0395) | (0.0334) |
| *Size* | 0.3215 *** | 0.1181 *** | 0.2845 *** | 0.3247 *** | 0.2723 *** |
| | (0.0412) | (0.0209) | (0.0376) | (0.0445) | (0.0352) |
| *Mature* | −0.0786 | −0.0847 | −0.0050 | −0.0686 | 0.0618 |
| | (0.1083) | (0.0714) | (0.1229) | (0.1072) | (0.0931) |
| *Lev* | −0.1715 ** | −0.0194 | −0.0828 | −0.1718 * | −0.1082 * |
| | (0.0842) | (0.0645) | (0.1114) | (0.0895) | (0.0610) |
| *Share* | −0.3208 * | −0.0784 | −0.2581 | −0.3424 * | −0.2550 |
| | (0.1863) | (0.1035) | (0.1935) | (0.1873) | (0.1677) |
| *lnManager* | −0.0162 | 0.0471 | −0.0561 | −0.0153 | −0.0128 |
| | (0.0489) | (0.0475) | (0.0564) | (0.0509) | (0.0415) |
| *lnGov* | 0.4261 *** | 0.2899 *** | 0.3222 *** | 0.4249 *** | 0.3940 *** |
| | (0.0586) | (0.0426) | (0.0469) | (0.0608) | (0.0496) |
| Firm FE | Yes | Yes | Yes | Yes | Yes |
| Year FE | Yes | Yes | Yes | Yes | Yes |
| Constant | −0.2853 | −0.1367 | −0.2456 | −0.2866 | −0.5902 ** |
| | (0.3339) | (0.2401) | (0.3969) | (0.3212) | (0.2870) |
| *N* | 9571 | 9642 | 9642 | 8891 | 15449 |
| $R^2$ | 0.6782 | 0.5603 | 0.6734 | 0.6766 | 0.6666 |

Note: Robust standard errors for clustering to the city level are in parentheses. The *, **, and *** indicate significance at the 10%, 5%, and 1% levels, respectively. Unless otherwise indicated, see Table 3.

4.3.2. Alternative-Dependent Variables

Referencing Lee and Nie [49], we introduce two alternative variables to measure firms' green technology innovation. (a) The number of green invention patents granted to firms that year (*Ginvent_g*) and (b) the number of green utility model patent applications

to firms that year (*Gutility*). In the specific calculation, the number of patents in the two categories is added to 1 to take the natural logarithm and reduce heteroskedasticity. In Table 4, the results reported in Columns (2) and (3) indicate that the positive impact of the Chinese IPR model city pilot policy remains significant. Moreover, combined with the baseline regression results, it can be seen that the pilot policy mainly promoted the green invention applications by enterprises (0.1485) and had a minor impact on green utility model patents (0.0843).

### 4.3.3. Change the Research Sample

First, considering that some of the enterprises in the original sample have never applied for green patents, if most of such enterprises are concentrated in nonpilot cities, this may affect the regression results. Enterprises whose green patents have always been zero are excluded in this study to eliminate this interference. These regression results are shown in Column (4) of Table 4. Second, to enhance the credibility of the results, we have used an all-industry sample that does not include the financial category. The results in Column (5) reveal that the regression coefficients of *Treat × Post* are significantly positive at the 1% level for the selected sub-sample and the expanded sample size.

### 4.3.4. Placebo Test

We performed a placebo test by randomly selecting the treatment group from the sample. First, we used Stata 15.1 software to randomly select 492 of the 849 sample firms as the treatment group, and the policy time was also randomly given. Subsequently, the process was repeated 1000 times to obtain 1000 sets of policy dummy variables ($Treat^{random} \times Post^{random}$). Finally, the kernel density and *p*-value distributions of $\beta^{random}$ are presented in Figure 2. We find that the estimated coefficients $\beta^{random}$ are concentrated around zero, which is much smaller than the actual estimate of 0.1485. Most *p*-values were also above 0.1 and statistically insignificant. This demonstrates that omitted variables do not impact the estimation results and baseline regression results are robust.

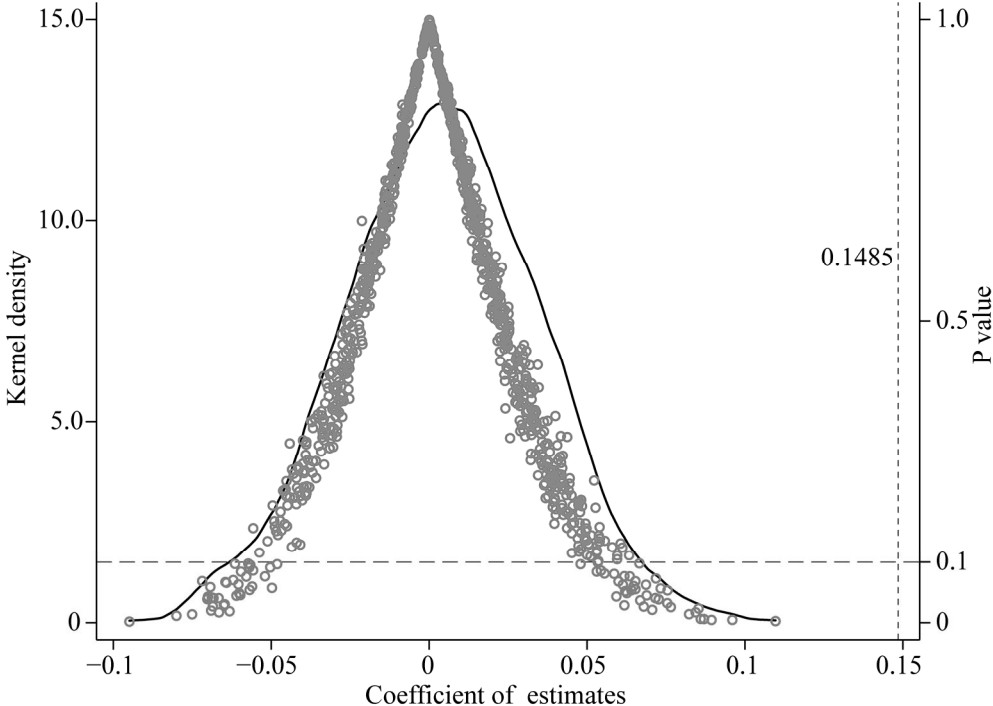

**Figure 2.** Placebo test. Note: The right-hand vertical line is the estimated coefficient of 0.1485.

## 5. Heterogeneity Analysis and Mechanism Verification

### 5.1. Heterogeneous Effects

The heterogeneity test aims to clarify whether the promotion effect of the Chinese IPR model city pilot policy on green innovation differs across firm size, nature of property rights, and industry characteristics. Accordingly, we perform heterogeneity analysis by adding interaction terms of policy dummy variables with firm characteristics dummy variables based on model (1).

$$\text{Ginvent}_{it} = \alpha + \beta Treat_i \times Post_t \times Group_j + \sum \gamma_j X_{it} + \delta_i + \theta_t + \varepsilon_{it} \tag{3}$$

where $Group = \{Scale, SOE, Reverse\}$ are dummy variables, $Scale = 1$ when the sample firm is large and 0 otherwise; $SOE = 1$ when the firm is state-owned and 0 otherwise. Similarly, $Reverse = 1$ when the company is in the reverse technology industries and 0 otherwise. What we are interested in is the estimated coefficient $\beta_1$ of $Treat_i \times Post_t \times Group_j$.

#### 5.1.1. Heterogeneous Effects by Firm Scale

We classify firms into large firms (equal to and above the sample median) and small firms (below the median) based on their assets. The estimation results are shown in Column (1) of Table 5. The coefficient of $Treat \times Post \times Scale$ is significantly positive. It implies that the Chinese IPR model city pilot policy has a larger effect on large firms' green innovation. One of the possible reasons is that IP protection provides institutional security for large firms to capture monopoly profits in a particular technology area by reducing R&D spillovers and the risk of imitation by external competitors [11]. Large firms may be more inclined to develop and adopt green technologies to maintain a higher market share and a competitive advantage. As highlighted by Khanra et al. [50], green innovation has the potential to serve as a crucial firm resource for building a competitive advantage while also contributing to sustainable development. Another reason lies in innovation resource constraints. Large firms have more capital, talent resources, and technology accumulation than small firms. As Eppinger et al. [23] state, larger firms have more resources to bring sustainability solutions to manufacturing. With strengthened IP protection, large firms can better leverage their financial and technological advantages, rationally allocate innovation resources, and achieve more green technological innovations. While policymakers have encouraged technological innovation by strengthening IP protection, small firms may also achieve limited results in the short term, facing green innovations of greater technological complexity [18,51].

#### 5.1.2. Heterogeneous Effects in Ownership

We divided the enterprises into state-owned and private enterprises to test whether the nature of enterprise ownership differentiates the policy effects of the Chinese IPR model city pilot. As seen in the results shown in Column (2) of Table 5, state-owned enterprises (SOEs) in the pilot cities have applied for more green patents than private enterprises. Inconsistent with previous research [52], the impact of IPR policies is more significant among SOEs. After implementing the pilot policy, the level of green technology innovation in SOEs is 18.33% higher than in private enterprises. One potential explanation is that SOEs possess a competitive edge over ordinary enterprises regarding access to capital lending and political connections. According to Wang et al. [53], SOEs possess more resources than non-SOEs, bolstering their robust performance in energy conservation, emission reduction, and social responsibility. SOEs enjoy an edge in securing bank credit and actively engage in high-quality green innovation. By implementing corporate entrepreneurship strategies, SOEs with political connections demonstrated an addition of roughly 0.076 green patent authorizations annually [54]. The Chinese IPR model city pilot policy further reinforces these advantages. Ang et al. [7] emphasized that bolstered enforcement of intellectual property rights diminishes information asymmetries, enhancing firms' access to external equity and debt financing and fostering R&D and innovation.

**Table 5.** Heterogeneity analysis.

| Variables | Firm Scale | Equity Nature | Industry Type |
|---|---|---|---|
| | (1) | (2) | (3) |
| | *Ginvent* | *Ginvent* | *Ginvent* |
| *Treat* × *Post* × *Scale* | 0.1998 *** | | |
| | (0.0440) | | |
| *Treat* × *Post* × *SOE* | | 0.1833 *** | |
| | | (0.0542) | |
| *Treat* × *Post* × *Reverse* | | | 0.2609 *** |
| | | | (0.0560) |
| *Size* | 0.2968 *** | 0.3273 *** | 0.3192 *** |
| | (0.0404) | (0.0417) | (0.0421) |
| *Mature* | −0.0280 | −0.0030 | −0.0872 |
| | (0.1083) | (0.1035) | (0.1056) |
| *Lev* | −0.1731 ** | −0.1612 * | −0.1807 ** |
| | (0.0848) | (0.0849) | (0.0843) |
| *Share* | −0.3355 * | −0.3646 ** | −0.3090 * |
| | (0.1838) | (0.1801) | (0.1837) |
| *lnManager* | −0.0211 | −0.0223 | −0.0188 |
| | (0.0479) | (0.0479) | (0.0478) |
| *lnGov* | 0.4039 *** | 0.4211 *** | 0.4209 *** |
| | (0.0577) | (0.0587) | (0.0599) |
| Firm FE | Yes | Yes | Yes |
| Year FE | Yes | Yes | Yes |
| Constant | −0.2954 | −0.4528 | −0.2280 |
| | (0.3443) | (0.3299) | (0.3355) |
| *N* | 9642 | 9642 | 9642 |
| $R^2$ | 0.6833 | 0.6821 | 0.6832 |

Note: Robust standard errors for clustering to the city level are in parentheses. The *, **, and *** indicate significance at the 10%, 5%, and 1% levels, respectively. Unless otherwise indicated, see Table 3.

5.1.3. Heterogeneous Effects of Industry Characteristics

When a company's innovative product or R&D technology can be easily imitated or stolen by competitors, a company tends to reduce its R&D investment [12]. Especially in machinery and equipment manufacturing, where technology is easy to reverse engineer or secrecy is ineffective, companies are more likely to apply for patents to reduce R&D spillover through legal forms to protect their innovations. Based on surveys conducted by Moser [13], innovations in manufacturing machinery, agricultural machinery, and engines were more likely to be patented, while chemical innovations were rarely patented in Britain and the United States in 1981. The argument posits that chemical innovations were difficult to reverse engineer and could be protected through trade secrecy. Conversely, innovations in manufacturing machinery were deemed easier to replicate and relied heavily on patents for protection. A high frequency of patent infringement and a lengthy R&D process will push up the cost of innovation, leading firms in the above industries to be more sensitive to patent protection policies [55]. Therefore, we define industries like equipment manufacturing as reverse technology industries. The results in Columns (3) of Table 5 show that the coefficient of *Treat* × *Post* × *Reverse* is significantly positive. This finding reveals a higher green innovation level in reverse technology industries compared to other sectors after introducing the Chinese IPR model city pilot policy. Companies in industries where technology is easy to reverse engineer are more willing to patent their inventions to extend the life of their technology protection and reap the total rewards of innovation.

*5.2. Mechanism Verification*

According to the literature review and influence mechanism analysis, the Chinese IPR model city pilot policy improves companies' green technology innovation through two channels: promoting collaborative innovation and improving the human resource

structure of enterprises. To test the before mentioned impact mechanisms, we construct the following mediating effects model by referring to the approach of Baron and Kenny [56]:

$$Ginvent_{it} = \alpha_0 + \alpha_1 Treat_i \times Post_t + \sum \gamma_j X_{it} + \delta_i + \theta_t + \varepsilon_{it} \tag{4}$$

$$M_{it} = \beta_0 + \beta_1 Treat_i \times Post_t + \sum \eta_j X_{it} + \delta_i + \theta_t + \varepsilon_{it} \tag{5}$$

$$Ginvent_{it} = \varphi_0 + \varphi_1 Treat_i \times Post_t + \varphi_2 M_{it} + \sum \phi_j X_{it} + \delta_i + \theta_t + \varepsilon_{it} \tag{6}$$

where $M_{it}$ present the mediate variables, including cooperation and human capital. The total effect of the policy is given by $\alpha_1$, the direct effect is given by $\varphi_1$, and $\beta_1\varphi_2$ gives the indirect effect of the mediating variable. The previous baseline regression $\alpha_1$ is significantly positive. If $\beta_1$ and $\varphi_1$ are significant and the coefficient $\varphi_1$ is smaller than $\alpha_1$, then $M_{it}$ is a "partially mediated" variable, according to the tests of mediation.

### 5.2.1. Mediating Effects of Cooperation

As previously mentioned, R&D collaboration is one of the determining factors in driving companies' green technology innovation. Therefore, we selected two proxy variables to determine the mediating effect of open innovation. The first is a dummy variable for R&D collaboration (*Cooperate*1), which equals one if there is a joint application for a green patent and 0 otherwise. The second one is the logarithm of the number of joint patent applications (*Cooperate*2). According to the test process, the regression coefficient of *Treat × Post* in Column (1) of Table 6, the baseline regression result, is significantly positive, indicating that the Chinese IPR model city pilot policy can promote green innovation. Furthermore, the coefficients in Columns (2) and (4) are both significantly positive. Finally, the proxy variables for open innovation are significantly positive in Columns (3) and (5), indicating that the higher the firms' participation in R&D collaboration is, the higher the green innovation level is.

**Table 6.** Mechanism verification.

| Variables | (1) | Open Innovation | | | | Human Capital | |
|---|---|---|---|---|---|---|---|
| | | (2) | (3) | (4) | (5) | (6) | (7) |
| | *Ginvent* | *Cooperate*1 | *Ginvent* | *Cooperate*2 | *Ginvent* | *Tech* | *Ginvent* |
| *Treat × Post* | 0.1485 *** | 0.0343 ** | 0.1239 *** | 0.0639 *** | 0.1084 *** | 0.7734 ** | 0.1441 *** |
| | (0.0389) | (0.0136) | (0.0374) | (0.0226) | (0.0381) | (0.3648) | (0.0393) |
| *Cooperate*1 | | | 0.7183 *** | | | | |
| | | | (0.0438) | | | | |
| *Cooperate*2 | | | | | 0.6271 *** | | |
| | | | | | (0.0485) | | |
| *Tech* | | | | | | | 0.0054 *** |
| | | | | | | | (0.0017) |
| *Size* | 0.3259 *** | 0.0614 *** | 0.2818 *** | 0.0819 *** | 0.2745 *** | 0.0350 | 0.3277 *** |
| | (0.0412) | (0.0118) | (0.0414) | (0.0195) | (0.0434) | (0.3710) | (0.0418) |
| *Mature* | −0.0686 | −0.0574 | −0.0274 | −0.1054 | −0.0025 | −3.2632 ** | −0.0424 |
| | (0.1058) | (0.0455) | (0.1081) | (0.0880) | (0.1136) | (1.4810) | (0.1077) |
| *Lev* | −0.1811 ** | −0.0655 ** | −0.1341 * | −0.0919 * | −0.1235 | −0.6784 | −0.1801 ** |
| | (0.0837) | (0.0311) | (0.0783) | (0.0542) | (0.0775) | (1.1249) | (0.0839) |
| *Share* | −0.3245 * | −0.1729 ** | −0.2002 | −0.1671 * | −0.2197 | −5.2562 ** | −0.2740 |
| | (0.1846) | (0.0669) | (0.1718) | (0.0970) | (0.1710) | (2.4590) | (0.1875) |
| *lnManager* | −0.0166 | −0.0210 | −0.0016 | −0.0304 | 0.0024 | 0.3874 | −0.0239 |
| | (0.0486) | (0.0177) | (0.0437) | (0.0292) | (0.0416) | (0.5472) | (0.0497) |
| *lnGov* | 0.4246 *** | 0.0929 *** | 0.3579 *** | 0.1762 *** | 0.3141 *** | 1.0528 ** | 0.4174 *** |
| | (0.0582) | (0.0234) | (0.0534) | (0.0406) | (0.0545) | (0.4792) | (0.0586) |
| Year FE | Yes | Yes | Yes | Yes | Yes | Yes | Yes |
| Firm FE | Yes | Yes | Yes | Yes | Yes | Yes | Yes |
| Constant | −0.3223 | 0.1355 | −0.4197 | 0.2103 | −0.4543 | 25.8941 *** | −0.4874 |
| | (0.3296) | (0.1241) | (0.3310) | (0.2222) | (0.3240) | (4.4361) | (0.3421) |
| $R^2$ | 9642 | 9642 | 9642 | 9642 | 9642 | 9562 | 9562 |
| $N$ | 0.6824 | 0.3748 | 0.7171 | 0.4736 | 0.7262 | 0.7735 | 0.6829 |
| Sobel test | | | 2.3840 ** | | 2.6320 ** | | 1.6720 * |
| Proportion | | | 0.1660 | | 0.2700 | | 0.0280 |

Notes: Robust standard errors for clustering to the city level are in parentheses. The *, **, and *** indicate significance at the 10%, 5%, and 1% levels, respectively. Unless otherwise indicated, see Table 3. The Sobel test reports z-values and the corresponding asterisks also indicate the significance levels.

To further ensure significance, Sobel–Goodman mediation tests were also conducted. The results show that the statistics, Sobel z-values, are all larger than 0.97, indicating that

the mediating effect of open innovation is significantly present. The proportions of the mediating effect to the total effect are 16.6% and 27.0%, respectively.

### 5.2.2. Mediating Effects of Human Capital

A study by Barbieri et al. [21] shows that green patents are significantly higher than non-green patents in terms of the number of applicants, scope of protection, and originality. In particular, green patents received 31.8% more citations from subsequent inventions than non-green counterparts. This result implies that green technology innovation is more innovative and complex than general technology. Due to more technological components and unique knowledge sets, companies need to depend on a highly skilled workforce to develop green technologies. To test the mediating effect of human capital, we collected the company's employee information from the RESSET database. We manually grouped the company's employees' occupations into technicians and others. The former includes technicians and R&D personnel, mainly engineers, programmers, and laboratory researchers, engaged in technical work. We obtained a new dataset after excluding certain sample firms with missing data. Using this dataset, we calculated the share of scientific and technical workers in total employees (*Tech*), a mediating variable in the above model for regression estimation. As emphasized by Autor et al. [57], scientific and technical workers are involved in nonroutine and nonrepetitive tasks, which are not easily replaced by machine technology. Consequently, based on Autor et al. [57], we classify scientific and technical personnel as highly skilled labor. An increase in the proportion of scientific and technological personnel signifies an improvement in the skill composition of the firm's human capital.

The process of testing here is consistent with that described above. In Table 6, Column (6) results show that the Chinese IPR model city pilot policy significantly improves the firms' human capital structure by increasing the proportion of scientific and technological personnel. Combining the regression results in Column (7), it is clear that the pilot policy promotes green innovation by improving listed companies' internal human capital structure. The mediating effect under this path accounts for about 2.80% of the total effect.

## 6. Conclusions and Policy Implications

Implementing an innovation-driven development strategy and building a country with IPR fosters high-quality development and improves national economic competitiveness. In this context, we construct a progressive DID model with the listed manufacturing firms' panel data to systematically assess how IPR protection policies affect green technology innovation. The empirical evidence shows that the Chinese IPR model city pilot policy significantly improved the level of green innovation among the listed companies. The heterogeneity analyses show that policy treatment effects are more prominent for large firms, SOEs and reverse technology industries. The mechanistic tests reveal that the pilot policy encourages firms' R&D cooperation and increases human capital levels, positively impacting green technology innovation, and the contribution of R&D collaboration is relatively significant.

Based on the above findings, we propose the following policy implications. Firstly, policymakers should improve the policy system to protect green technologies' intellectual property rights, reduce technology imitation, piracy, and other infringements, and create a favorable environment for innovation. Secondly, the government should encourage industry leaders such as large firms and state-owned enterprises to strengthen further R&D cooperation with universities, research institutes, and other units to accelerate breakthroughs in green technology innovation. Thirdly, the government should join forces with universities, research institutes, and firms to cultivate green technology innovation professionals and high-quality technical and skilled personnel jointly. Furthermore, firms aiming to enhance market competitiveness and achieve technological breakthroughs must consistently assimilate advanced external knowledge and technology. A proficient level of human capital is vital in fostering the assimilation and understanding of state-of-the-art technological advancements. Consequently, it is imperative for firms to actively engage

in R&D collaborations within scientific research initiatives and prioritize the training and advancement of core technical personnel.

The paper exclusively delves into the impact of bolstering intellectual property protection on green technology innovation within China, thus imposing regional constraints on the study's findings. To advance the depth of this inquiry, we suggest that future research could expand in several areas. Firstly, it could investigate whether patent protection policies encourage green innovation among firms in other countries. Subsequent research might utilize cross-country data to conduct comparative studies, employing methods such as the one proposed by Ginarte and Park [58] to gauge the level of patent protection in each country and explore variations in the impact of patent protection on green innovation across different nations. Secondly, the mechanisms through which patent protection influences firms' green innovation need further exploration. While this paper argues for the mediating role of open innovation and improvements in human capital structure, it is essential to consider whether other variables act as mediators. For instance, further examination could test whether intellectual property protection accelerates green innovation by enhancing firms' technology absorption and learning capabilities.

**Author Contributions:** Conceptualization, D.C.; Methodology, D.C.; Software, D.C.; Validation, D.C.; Formal analysis, D.C.; Data curation, D.C.; Writing—original draft, D.C.; Writing—review & editing, S.C.; Supervision, S.C.; Project administration, S.C.; Funding acquisition, S.C. All authors have read and agreed to the published version of the manuscript.

**Funding:** This research was funded by the National Social Science Fund of China grant number [21XJY006].

**Institutional Review Board Statement:** Not applicable.

**Informed Consent Statement:** Not applicable.

**Data Availability Statement:** The data presented in this study are available on request from the corresponding author.

**Conflicts of Interest:** The authors declare no conflict of interest.

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
