# Peer review of "Patent Protection Policy and Firms’ Green Technology Innovation: Mediating Roles of Open Innovation and Human Capital"

_sustainability, doi:10.3390/su16052217_

Round 1

Reviewer 1 Report

Comments and Suggestions for Authors

It is an interesting Paper, but it lacks clearly stated research objective that should be presented in Introduction. A study is based on proper data (panel data from 849 listed manufacturing firms in China between 2007 and 2019), and research methods were selected properly.  The Paper focuses on an important research problem and it is potentially interesting for Sustainability readers.

Author Response

We sincerely appreciate your insightful and constructive comments on our paper. Following your comments and suggestions, we have revised the article to address all concerns. Below, we present point-by-point responses, quoting your comments in italics and responding in the normal font in the subsequent paragraphs. Also, we marked all the changes in red font in the Revised Manuscript. Please see the attachment.

Reviewer 2 Report

Comments and Suggestions for Authors

1. Introduction

In this section, the methodology of the research itself is mentioned, please transfer some sentences to the methodology section.

Literature Review

It is absent

If the authors decide to finish writing this chapter, I recommend that they also address the issue of sustainability. In this way, your contribution will correspond more thematically with the focus of the magazine

2. Research Background and Hypothesis

This part is suitable for introduction

2.1. The Chinese IPR Model City Pilot Policy –rather suitable for the introduction

2.2. IPR Protection and Innovation

Rework, add additional resources

2.3. Effect Mechanisms between IPR Protection and Green Innovation

2.3.1. Open Innovation

This chapter is off topic

2.3.2. Human Resources

Insufficiently elaborated part

3. Empirical Strategy and Data 191

3.1. Methodology, 3.2.  Variable Measurment  3.2. 1. Dependent Variable

Inadequately described research problem, neither the context nor the goal nor the problem of the research are clear.

3.3. Data

The text part is not related to the results of the work, which are presented in Tables 1 and 2.

4. Empirical Results

4.1. Time Trend Tests and Dynamic Effects

The text part is not related to the results of the work, which are presented in Tables 1 and 2.

4.2. Baseline Regression Results

I agree with the content of this section. I have no reservations about her.

4.3. Robustness Checks

4.3.1. PSM-DID Estimation

Research is non-identifiable.

6. Conclusions and Policy Implications

Rework

References

The authors used little specialized literature in their scientific work.

Author Response

(The authors gave the same response as above.)

Reviewer 3 Report

Comments and Suggestions for Authors

This study analyzes the impact of patent protection policies on firms' green technology innovation using panel data from firms. However, research gaps and the need for research, especially academic research, are poorly presented. In many parts, the evidence to support the claims is poorly presented. To improve the current manuscript, I point out the following items regarding the current structure and logic of the document. My detailed comments are attached.

Comments on the Quality of English Language

Moderate editing of English language required.

Author Response

(The authors gave the same response as above.)

Round 2

Reviewer 2 Report

Comments and Suggestions for Authors

I wish you a good day.

 Well thank you.

I agree with your edits and I recommend you publish the post.

Author Response

We want to express our great appreciation for your comments on our manuscript. Thank you.

Reviewer 3 Report

Comments and Suggestions for Authors

2nd Review for "Patent protection policy and firms' green technology innovation: Mediating roles of open innovation and human capital."

I have reviewed amendments to the manuscript following my feedback, for which I am grateful. However, a particular aspect remains unclear and needs further clarification.

Response 7 "5. (2.2. IPR Protection and Innovation) Before H1, the logic leading to H1 derivation from the characteristics of the Chinese IPR model city pilot policy is insufficient. In addition, the results of existing studies similar to H1 should be presented here.": The issues I pointed out have not yet been resolved.

Response 8: You have indicated that "The mechanism by which intellectual property protection affects firms' green innovation is different from all innovation". However, the manuscript does not fully explain how the mechanisms by which IP protection affects green innovation differ from all innovations. We believe it is essential to make this distinction explicit within the text to ensure the comprehensiveness of the manuscript and to avoid potential misunderstandings about the impact of IP protection on different forms of innovation.

Response 19: There is still a lack of explanation for why large firms are more likely to develop and adopt green technologies than small firms.

Author Response

We sincerely appreciate your insightful and constructive comments on our paper. Following your comments and suggestions, we have revised the article to address all concerns. Please see the attachment.
